# Noninvasive Detection of Bacterial Infection in Children Using Piezoelectric E-Nose

**DOI:** 10.3390/s22218496

**Published:** 2022-11-04

**Authors:** Tatiana Kuchmenko, Daria Menzhulina, Anastasiia Shuba

**Affiliations:** 1Department of Physical and Analytical Chemistry, Voronezh State University of Engineering Technologies, Voronezh 394000, Russia; 2Propaedeutics of Childhood Diseases and Polyclinic Pediatrics, Voronezh State Medical University Named after N. N. Burdenko, Voronezh 394000, Russia

**Keywords:** piezoelectric sensor, microbalance, volatile organic compounds, biomarkers, urine, bacterial infection, chemometrics, infection indicator

## Abstract

Currently, antibiotics are often prescribed to children without reason due to the inability to quickly establish the presence of a bacterial etiology of the disease. One way to obtain additional diagnostic information quickly is to study the volatile metabolome of biosamples using arrays of sensors. The goal of this work was to assess the possibility of using an array of chemical sensors with various sensitive coatings to determine the presence of a bacterial infection in children by analyzing the equilibrium gas phase (EGP) of urine samples. The EGP of 90 urine samples from children with and without a bacterial infection (urinary tract infection, soft tissue infection) was studied on the “MAG-8” device with seven piezoelectric sensors in a hospital. General urine analysis with sediment microscopy was performed using a Uriscan Pro analyzer and using an Olympus CX31 microscope. After surgical removal of the source of inflammation, the microbiological studies of the biomaterial were performed to determine the presence and type of the pathogen. The most informative output data of an array of sensors have been established for diagnosing bacterial pathology. Regression models were built to predict the presence of a bacterial infection in children with an error of no more than 15%. An indicator of infection is proposed to predict the presence of a bacterial infection in children with a high sensitivity of 96%.

## 1. Introduction

It is currently quite common that antibiotics are prescribed to children, even when there is no clear indication that a bacterial etiology is involved. This is because there is no diagnostic tool that can quickly determine the presence of microorganisms and their sensitivity to various groups of antibiotics. This is especially true for surgical patients, for whom the use of antibiotics does not always prevent infection [1]. In some cases, antibiotics are not recommended for surgery [2]. Abdominal infections, however, need broad-spectrum antibiotics [3]. Their use is often insufficient to prevent the development of bacterial complications or is accompanied by side effects [4,5]. It is known that long-term use of antibiotics can reduce the risk of re-infection, although it can also cause addiction and adaptation of microorganisms to them [6]. This, in turn, further leads to the need to use groups of reserved drugs that are stronger; however, the list of complications and contraindications is much wider, and many have an age limit. Furthermore, studies indicate that the use of antibiotics to prevent postoperative complications is not effective [7].

Therefore, in May 2014, the 67th WHO Assembly adopted a resolution to fight antimicrobial resistance [8]. It is especially important to consider the impact on the normal microbiota when treating children, since the increased resistance of the internal pathogenic microbiota to antibiotics can lead to the disruption of the normal microbiota of the body and cause digestive problems [9]. Therefore, it is vital to find out the etiology of the disease as early as possible to avoid unnecessary antibiotic therapy [10], and to use alternative methods to prevent surgical infections [11]. 

Blood and urine tests are common clinical tests used to diagnose pathological conditions. Recent research has focused on noninvasive methods for diagnosing disease, especially in cases where initial symptoms are vague. Blood sampling for analysis is very stressful for children, often accompanied by crying, screaming, and other negative emotions. In comparison, the collection of urine for analysis is a more physiological and atraumatic process. Moreover, the composition of urine reflects the condition of health of organs and systems in the human body [12]. It is possible to increase the information content of a general urinalysis by assessing the qualitative and quantitative composition of the gaseous fraction of urine. The volatile compounds of urine change more significantly along with the proteome during the development of the disease. Consequently, studies of the gas composition of urine were carried out [13,14], and methods for determining volatile biomarkers in urine were proposed [15,16]. Bacterial metabolites secreted on nutrient media have also been studied [17,18], including volatile compounds secreted during infection of tissues [17] and exudates [19], lesion by *Staphylococcus aureus* [20], respiratory and gastrointestinal diseases [21], bacteriuria [22,23,24]. A method for assessing the standard indicators of the general urine analysis and other diagnostic parameters by the volatile fraction of urine samples has been proposed [25,26]. The association of volatile compounds in urine with diseases of the intestine, kidney, cancer of various locations [26,27], diabetes [28], genetic diseases [29] has been shown. Differences were observed in urine samples from healthy humans when infected with SARS-CoV-2 [30] and Mycobacterium tuberculosis [31,32]. Moreover, the gas composition of urine in relation to other biological fluids and samples was investigated [33,34] to determine the characteristics of the metabolism. Particularly, volatile markers associated with acetylcholine metabolism have been identified as a change in the urinary proteome [35]. A change in the urinary proteome was also established during infection of the abdominal cavity with *Escherichia coli* and *Staphylococcus aureus.* It was shown that their combined presence causes significant differences from a monoinfection [36]. Based on these findings, it may be possible to assess the effect of an infectious agent on the composition of urine, including volatile metabolites.

When analyzing the output curves of sensors to predict the presence of an infection, various data processing methods are used, both with a preliminary selection of variables and the entire original data, such as projection methods for data compression and decomposition [26,37,38], including Internet of Things sensor systems [39]. The most commonly used regression approaches (partial least squares regression, linear cross-correlation technique, black box model) are applied to the output data of arrays of sensors and gas chromatography. The new approaches were proposed to reduce the input data dimension and to estimate the predicted indicators [40].

The purpose of the work is to evaluate the possibility of using an array of chemical sensors with various sensitive coatings to determine the presence of a bacterial infection in children by analyzing the equilibrium gas phase of urine samples.

## 2. Materials and Methods

### 2.1. Collection of Biosamples and Study Design

To determine differences in the composition of the equilibrium gas phase (EGP) in urine samples from patients aged 1 to 16 years with bacterial contamination of the urogenital tract or inflammatory processes of other tissues and organs of bacterial etiology, 90 urine samples from patients from different departments of a children’s hospital were examined. Additionally, if surgical intervention was necessary (furuncle, abscess, etc.), after removal of the inflammation source, bacteriological investigation of the biomaterial was performed in 33 patients to determine the presence and type of pathogen. The studies were conducted based on the clinical laboratory of Region Hospital No. 2 in Voronezh, in accordance with the voluntary consent of patients or their legal representatives, over the period 2017–2018. During the experiment, contact with patients was not carried out; special permission from the ethical committee was not required.

#### 2.1.1. Urine Sample Analysis

Urine sampling was carried out in the morning on an empty stomach in accordance with the rules for sampling for bacterial culture at various stages of treatment. In the clinical diagnostic laboratory of the hospital, urine samples from each patient were divided into three portions. One portion (volume 20 cm^3^) was placed in a sealed bottle (volume 50 cm^3^) with a propylene cap and kept at room temperature for 15 min, after which the equilibrium gas phase was analyzed. The second portion was examined on the Uriscan Pro analyzer (manufactured by YD Diagnostics, South Korea) with URISCAN 11 test strips to determine levels of 11 analytes. The third portion was taken for examination of the sediment on an Olympus CX31 microscope (Japan) using slide plates (Lachema, Czech Republic). The indicators for general urine analysis (GUA) and sediment microscopy were determined in accordance with standard methods and recommendations [41,42]. Some indicators of GUA are presented in Table 1. To indicate semi-quantitative indicators of a general urine test, the following designations were used: ~0—absent in the field of view, +—an insignificant amount was present in the sample, ++—a moderate amount, +++—covered almost the entire field of view (Table 1).

The results of a complete analysis of urine samples by standard parameters are presented in Appendix A.

#### 2.1.2. Biomaterial Analysis

During surgery, biomaterial was taken from the source of the inflammation according to the recommendations [43]. The biomaterial was analyzed by the hospital staff in the clinical diagnostic laboratory using cultural, microbiological and biochemical methods in accordance with the methodological recommendations [44].

### 2.2. Analysis of Volatile Compounds

#### 2.2.1. Device and Sensor Array 

Analysis of the equilibrium gas phase of urine samples was carried out on the MAG-8 gas analyzer—piezoelectric nose (LLC “Sensorika–Novye Technologii”, Russia) [45] (Figure 1). The detection cell of the device was equipped with a set of piezoelectric sensors coated with films: polyethylene glycol sebacate (PEGSb—sensor 1), triton X-100 (TX-100—sensor 2), dicyclohexane-18-crown-6 (18C6—sensor 3), polyoxyethylene sorbitan monooleate (Tween—sensor 4), methyl red (MR—sensor 5), bromocresol blue (BCB—sensor 6), multiwalled carbon nanotubes (MCNT—sensor 7). Piezoelectric quartz resonators with initial frequency of oscillation 10.0 MHz (LLC “Piezo”, Moscow, Russia) were unsoldered. The electrodes of the resonator were degreased by organic solvents (acetone or chloroform). The sorbents without preliminary preparation were dissolved in appropriate solvent (acetone—for PEGSb, TX-100, MR, toluene—for 18C6, Tween, ethanol—for BCB, chloroform—for MCNT) with concentration of 5 mg/mL. Then these solutions were used to form the sensitive coating. Coatings on sensors were formed by drop casting (sensors 1–4) or dip-coating (sensors 5–7) according to the previously described techniques [46].

The choice of sensors for an array is influenced by their high sensitivity to various classes of volatile substances, including volatile biomarkers of diseases in the urine [14,15,16]. Films of 18C6, Tween were chosen for the detection of carboxylic and hydroxy acids [47,48], and MCNT, BCB, MR for ammonia and amines [49,50,51]. PEGSb was selected for detection of acids, alcohols, ketones [52,53], and TX-100 for nitrogen- and sulfur-containing compounds [54,55]. Moreover, the selected films are stable for at least a year when analyzing small amounts of volatile substances [53]. Therefore, the resulting array should be effective for solving the problem. The mass sensitivities of selected sensors to some VOC vapors, which were obtained in earlier studies [47,48,49,50,51,52,53,54,55,56], are presented in Table 2. 

Before starting research in the hospital, the array of sensors was trained on a set of volatile compounds: ethanol, butanol-1, acetone, acetic acid, butyric acid, valeric acid, isovaleric acid, ammonia, diethylamine, piperidine, hydrogen sulfide (from ferrous sulfide and hydrochloric acid), phenol, ethyl acetate, dimethylacetal dimethylformamide (classification puriss., LLC “OS Reachem”, Moscow, Russia). The relative standard deviations of sensor signals for the studied volatile compounds are presented in Table 2. The limit of detection for the volatile compounds using the selected sensors was between 0.012 and 135 mg/m^3^, as estimated earlier [48,53,56].

In the software of the device, the signals of the sensors (∆*F*_i_, Hz) were recorded in time as chronofrequency grams. These signals were then analyzed to find the maximum sensor signal (∆*F*_max,i_), the area of a “visual print” of the maximum sensor signal (S_max_), the kinetic “visual print” of the entire sensor signal (S_sum_), and for each sensor (S_i_), the kinetic “visual print” of the time mask for the entire array (S_sumMK_) and for each sensor (S_MK,i_). These numbers were calculated automatically by the software [57]. The description of the basic version of the software was presented in [58]. The description of kinetic “visual prints” and the principle of choosing a time mask were described in [46,59].

#### 2.2.2. Technique of Measurement

The general technique of gas phase measurement over a urine sample can be presented as a scheme in Figure 2.

With a sterile syringe, 5.0 cm^3^ of the equilibrium gas phase was taken from the sampler over the urine sample and injected into the detection cell of the device. At the same time, the measurement record was turned on in the software. The time for measuring the sorption of volatile compounds of a urine sample was 2 min. Then, the measurement was saved to the database, and the detection cell was purged with dehumidified air, until the initial oscillation frequency of the sensor was restored (corresponding to the oscillation frequency of the piezoelectric resonator with a selective coating before the start of measuring urine samples). The restoration of the initial oscillation frequency of sensors *(F*_0_, Hz) was a criterion for the degree of regeneration of sensor coatings to ensure metrological measurement reliability, as shown earlier [32,33]. The restoration of the sorption layers was achieved within 1 min after the measurement under hospital conditions; therefore, the total measurement time for one sample was 3 min.

#### 2.2.3. Sensor Data Processing 

The analytical signals of the sensors and chronofrequency grams were used to calculate additional parameters of the sorption of volatile compounds of urine samples: the parameters of the efficiency A_i/j_ and the stability *γ_i_* of sorption and the geometry of “visual prints” (*m*_ijn_*, α*_ijn_) according to the Equations [56,59]:A*_i/j_* = ∆*F*_max,*i*_/∆*F*_max,*j*_,(1)
γ_i_ = ∆F_i(5)_/∆F_i(60)_,(2)
(3)$$m_{\mathrm{ijn}}=\frac{\sqrt{\Delta F_{\max,i}^{2}+\Delta F_{\max,j}^{2}-\Delta F_{\max,i}\cdot \Delta F_{\max,j}\cdot \sqrt{2}}}{\sqrt{\Delta F_{\max,j}^{2}+\Delta F_{\max,n}^{2}-\Delta F_{\max,j}\cdot \Delta F_{\max,n}\cdot \sqrt{2}}}$$
(4)$$\begin{array}{cc} \alpha_{\mathrm{ijn}} & =\arcsin\left(\frac{\Delta F_{\max,i}\cdot \sqrt{2}}{2\cdot \sqrt{\Delta F_{\max,i}^{2}+\Delta F_{\max,j}^{2}}-\Delta F_{\max,i}\cdot \Delta F_{\max,j}\cdot \sqrt{2}}\right)\\ & +\arcsin\left(\frac{\Delta F_{\max,n}\cdot \sqrt{2}}{2\cdot \sqrt{\Delta F_{\max,j}^{2}+\Delta F_{\max,n}^{2}}-\Delta F_{\max,j}\cdot \Delta F_{\max,n}\cdot \sqrt{2}}\right) \end{array}$$
where i, j, n—numbers of sensors in the array, the j-th sensor is between the i-th and n-th; ∆*F*_i*(5)*_, _(60)_—is the signal of the i-th piezoelectric sensor at the 5th and 60th second of the sorption of vapors of substances.

These parameters primarily reflect the qualitative composition of the equilibrium gas phase. Some of them are used to identify volatile compounds. More information about the use of sorption parameters can be found in [53,58,60]. Based on the calculated values of these parameters, volatile compounds were identified in the equilibrium gas phase of urine samples. The calculated values A_i/j_, *m*_ijn_ and *α*_ijn_ for urine samples were compared to the tabular values established earlier [59]. A substance was considered identified if the values for at least one calculated parameter A_i/j_, *m*_ijn_*, α*_ijn_ coincided with the table value of parameter within the coincidence criterion *d*.

The initial data for processing by the multivariate analysis method was formed from the following output data of the sensors: analytical signals (∆*F*_max,i_), areas of “visual prints” of maximal sensor signal (S_max_), kinetic “visual prints” of the entire sensor signals (S_sum_) and for each sensor (S_i_), kinetic “visual prints” of signals by the time mask for the entire array (S_sumMK_) and for each sensor (S_MK,i_), as well as the calculated parameters for all combinations of sensors in the array without repetition. In total, the initial data matrix was 122 × 80 in size. The data matrix was processed using the module for Microsoft Excel and Unscrambler X 10.0.1 (CamoSoftware AS, Oslo, Norway). The initial data were autoscaled before applying principal component analysis and multivariate regression methods. 

The method of partial least squares was used to build regression models. The full cross-validation method was chosen as the model validation algorithm. The training and test sets for building regression models were formed from the initial data matrix.

To predict the presence of urinary tract infections, 18 samples were taken as a training sample from patients from various departments. The value of the predictive factor was coded: “1”—the presence of bacteria in the sample (n = 5); “−1”—absence of bacteria (n = 13). As the initial set of variables, the output data from the array of sensors were taken, which were selected by principal component analysis. Further, variables with small regression coefficients and loadings for the first two principal components were excluded from this set until the minimum RMSE value was reached.

Thirteen samples from patients in the surgical departments were selected as a training sample to predict the presence of pathogenic microorganisms in the source of inflammation. The values of the predictive factor coded “1”—the presence of microorganisms (n = 8); “0”—absence of microorganisms (n = 5). The selection of variables for the model was carried out in the same way as in the previous model.

The sensitivity and specificity of the method for assessing the presence of a bacterial infection based on the results of the analysis of the EGP of urine samples were calculated as for variables with a binary response according to the Equations [61]:Sensitivity = N_CP_/(N_CP_ + N_FN_)(5)
Specificity = N_CN_/(N_CN_ + N_FP_)(6)
where N_CP_—number of correct positive results, N_FN_—number of false negative results, N_CN_—number of correct negative results, N_FP_—number of false positive results. 

## 3. Results

### 3.1. The Results of the General Analysis of Urine and Bacteriological Culture

The presence of urinary tract infections was established in 29 patients, and the presence of pathogenic microorganisms was found in 21 patients after surgery (Table 1). In half of the cases, the pathogenic microorganisms belonged to the genus *Staphylococcus*, moreover *Staphylococcus aureus* was found in 28% of cases. In the other cases, pathogenic *E. coli* and microorganisms of the genus *Streptococcus* were found. Furthermore, in 26% of cases, no pathogenic microorganisms were found in the biopsy by the bacteriological studies. Microscopy of urine sediment in half of the samples revealed the presence of mucus, which in 23% of cases was accompanied by leukocyturia.

### 3.2. The Results of the Identification of Volatile Substances in the Equilibrium Gas Phase of Urine Samples

The results of the identification of volatile substances in the equilibrium gas phase of urine samples by the parameters of sorption are presented in Table 3.

It should be noted that the presence of volatile compounds in the equilibrium gas phase above urine samples does not indicate a certain type of pathology. However, the presence of their combination can be used to judge the course of inflammatory processes, including bacterial etiology. The relationship between the numerical values of the identification parameters and clinical indicators is shown using multivariate data analysis.

### 3.3. Selection of Informative Sensor Output Data by Principal Component Analysis 

It has already been established that out of the entire set of calculated parameters for the selected array of sensors (98 parameters), only 32 were used to identify the substances. The most significant diagnostic information was extracted from the set of identification parameters using principal component analysis (PCA) (Figure 3). A satisfactory model with five principal components and an explained variance of 85% was obtained.

The first principal components were the most significant and could distinguish several groups of samples. According to the first principal component (horizontal axis), two groups of samples were distinguished: one was more dispersed (positive values), the second was more crowded (negative values).

Samples from the first group (negative values for PC-1) were aseptic, or samples from patients after a course of antibiotic therapy. Samples from the second group (positive values for PC-1) referred to pathologies with a bacterial infection, including against the background of taking antibiotics. It was also possible to separate samples located at the top (highlighted by the area, Figure 1), which were characterized by the presence of minor injuries and courses of physiotherapy, without taking antibiotics. The influence of individual parameters on the grouping of samples was evaluated according to the loading plot (Figure 4). 

At the next stage, PCA modeling of all output data of the array of sensors was carried out with full cross-validation. The model was characterized by overdetermination and a strong correlation of input variables. Therefore, insignificant and strongly correlated variables (low loadings > 0.1) were sequentially removed using the approach described in [48]. After optimization, a model was obtained (Figure 5) with a high correlation between the calibration and explained variances. The analytical signals of the sensors, the areas of kinetic “visual prints”, the parameters of sorption efficiency and stability were left as the optimal variables for modeling. There are several extreme points in Figure 5. For samples Nos. 15, 56, the presence of a source of inflammation before the start of surgical or therapeutic intervention was typical. For samples Nos. 62, 3, 2, in contrast, there were no or completely treated pathologies.

The microbiological contamination of urine samples was one of the main indicators affecting the distribution of samples on the scores plot.

### 3.4. Prediction of Bacterial Infection by Sensor Data Using Multivariate Partial Least Squares 

A model was built to predict the presence of bacteria in the urine using partial least squares regression. The set of output data from the array of sensors, which had previously been optimized by the method of principal components, was taken as variables.

It was found that two principal components were optimal with an explained variance of 64%, while the value of the RMSE was 0.56 (R2 = 0.64, slope = 0.292). On the scores plot (Figure 6), urine samples with microbiological contamination (red circle) and without it (blue square) are separated.

The output data of the sensors array indicated that the optimal variables for the prediction of microbiological contamination of urine samples were the signals from two sensors, the area of “visual print” of sensors S_MK,i_, parameters of sorption A_i/j_ and *γ*_i_. The most important variables for this model were the signal of the sensor with the MCNT film and the parameter A_6/7_ and *γ*_4_. Plots of regression coefficients for the PLS model are presented in Appendix A The correctness of the resulting model was checked by testing it on a new test set (Table 4).

It has been established that almost all samples were adequately described by the model, despite the significant values of the calculated deviations.

Further, a PLS model was built to predict the presence of pathogenic microorganisms in biomaterial from an inflammatory source. The same set of output data from the array of sensors was used as initial variables for modeling. This was carried out using principal component analysis. It has been established that the use of two principal components with an explained variance of 70% is optimal for modeling, the RMSE prediction error is 0.42 (R2 = 0.68, slope = 0.306). Two groups of samples can be distinguished on the scores plot (Figure 7).

In this case, the optimal set of variables includes the signals of four sensors, the area of “visual print” of sensors S_MK,i_, parameters of sorption A_i/j_ and *γ*_i_, and the most significant for the model are the analytical signal of the sensor and the area of the kinetic “visual print” by the time mask for the sensor with the MCNT film, the parameters A_3/5_, A_6/7_ and the parameter *γ*_6_. Plots of regression coefficients for the PLS model are presented in Appendix A The correctness of predicting the presence of microorganisms in inflammation source was checked with samples from the test set (Table 5).

The use of mathematical processing for the results of the analysis of urine samples by an array of sensors is very effective; however, it requires the availability of appropriate software and computing resources. Therefore, for faster data processing and obtaining screening information about the presence or absence of inflammatory processes and bacterial infection in the body, an infection indicator (*InfI*) is proposed, calculated by the formula:$$\mathrm{InfI}=m_{217}\cdot \alpha_{417}+m_{234}·A_{2/4}$$

This formula is an analogue of the formula for the scalar product of vectors, if we consider them in the plane of the identification parameters of the array of sensors. The maximum contribution to the ranking of samples according to the first principal component of the PCA model is the reason for the choice of precisely these identification parameters. The calculated values of *InfI* for all the studied samples are presented in Table 6.

The presence of inflammatory processes and bacterial infection in the body was established if *InfI* > 1.45. The value of the indicator was calculated according to the scores plot (Figure 3) as the ratio of the average values of the coordinates for the first principal component for two groups: samples with bacterial infections and samples with aseptic diseases. The obtained *InfI* values were compared with all clinical parameters of the samples. The presence of a bacterial infection and inflammation was established in accordance with the clinical diagnosis, the presence of a pathogen in the biomaterial, the presence of bacteria, mucus, or an increased content of protein and leukocytes in the urine. If these laboratory parameters were within the normal range and *InfI* ≤ 1.45, then this result was considered as correct negative; if a bacterial infection was diagnosed and *InfI* > 1.45, the result was considered as correct positive. 

Figure 8 presents general recommendations for using an array of sensors to increase the diagnostic information of urine analysis, considering previous studies [25].

The use of the obtained models according to this scheme enables a quick and efficient additional analysis of urine to be conducted without increasing the costs.

## 4. Discussion

Among the indicators of the general analysis of urine, the results of sediment microscopy are important. The presence of salts (oxalates, urates), mucus, protein, and erythrocytes in the samples will affect the redistribution of volatile substances at the gas–liquid interface, and, therefore, the results of analysis by an array of sensors. Microscopy of the urine sediment in half of the samples revealed the presence of mucus, which in some cases was accompanied by leukocyturia (more than five per field of view). This may indicate the presence of a nonspecific inflammatory process in the urogenital tract with other indicators within the normal range [62].

It is known that many types of pathologies are characterized by a certain odor and, consequently, by a set of volatile substances. Many substances that are metabolites in pathogenic processes are found in human secretions [14,15]. Therefore, volatile markers were identified in the EGP of urine samples according to identification parameters calculated by Equations (1)–(4). It has been established that ethanol, butanol, and their oxidation products (acetic and butyric acids) are present in almost all samples. The distribution of the presence of other substances in the EGP of urine samples is heterogeneous by department and can be determined both by the processes occurring in the body and by the peculiarities of the metabolism of drugs under the standard treatment protocol. It was noted that the presence of hydrogen sulfide in the EGP of the urine is indicative for injuries and inflammatory processes that do not violate the integrity of the skin, but require surgical intervention (furuncle, abscess, cyst, closed fractures with displacement) (surgical departments, Table 3). Hydrogen sulfide is identified in urine samples from patients who are characterized by the absence of pathogenic microorganisms during bacteriological studies of the biomaterial, which indicates the occurrence of an aseptic inflammatory process. The presence of acetone, ethyl acetate and isovaleric acid is typical for patients from the neurosurgical and purulent-septic departments. The results indicate that metabolites associated with inflammatory processes of bacterial origin are similar to those found in temporary metabolic disorders associated with head injuries. Phenol is more often present in the EGP of urine samples in neurosurgical pathologies (surgical departments, Table 3). The presence of aliphatic amines and ammonia in the EGP of urine samples from patients with purulent-septic pathologies and thermal injuries indicates a massive lesion. The presence of cyclic amines and acetals in EGP of urine samples is difficult to differentiate depending on the pathology, since they can be products of the metabolism of drugs used for treatment.

Upon performing the principal component analysis, it was found that the identification parameters for hydrogen sulfide, valeric and isovaleric acids (*m*_217_, *α*_417_) and ethanol and butanol (*m*_234_, $A_{2/4}$) could be used to select aseptic samples. The parameters for ethanol and butanol were found to be able to select samples with bacterial infection (Figure 4). For the selection of samples with minor injuries, the parameters $A_{4/7}$ and α_317_ are of the greatest importance; according to which, ethanol, butanol, acetone, and the absence of diethylamine were detected in the EGP of these samples. That means that not only is the presence of any marker substance important, but conversely, so is the absence of a specific marker or presence in a small concentration, below the detection limits for these parameters, for ranking samples into groups.

When comparing the scores and loading plots of the PCA model built by the entire set of output data of the array of sensors (Figure 5), it was found that to select samples corresponding to inflammatory conditions (samples No. 15, 56), the most important variables were the stability parameters (*γ*_i_) of sorption (signed in Figure 5). For evaluating the effectiveness of the treatment of bacterial infections (samples No. 2, 62), the primary data of the array of sensors had the greatest importance—the analytical signals of the sensors (∆*F*_max,i_) and the areas of kinetic “visual prints” by the time mask (S_sumMK_ и S_MK,i_) (shown in Figure 5).

It was found, upon analysis of the score plots of the PLS model for predicting the bacterial contamination of urine samples (Figure 6), that the group of samples without bacteria in the urine was more dispersed, which is associated with additional factors, such as the presence of mucus and salts in the urine, which affects the redistribution of volatile substances at the “gas–liquid” phase boundary. Therefore, it influences the results of analysis by an array of sensors. At the same time, the presence of mucus in the urine, as an indicator of inflammation, will significantly affect the prediction results, with the possibility of a false positive prediction.

According to the values of weighted regression coefficients, it was found that the most important and significant for predicting the microbiological contamination of urine samples were the output data of the sensors with films sensitive to amines and ammonia—BCB and MCNT (Appendix A).

It was found that almost all samples were adequately described by the model from the results of the prediction for the test set (Table 4). The best prediction results were achieved for samples in which other clinical indicators confirmed the presence of inflammation (presence of the protein, leukocytes or mucus). Overestimation of the results of the prediction, and as a result, false positive results, could be observed for urine samples with the presence of mucus and other compounds that cause the presence of turbidity. False-negative prognostic results were associated with the presence of erythrocytes or ketone bodies in the urine.

One sample was assigned to a different diagnostic group based on the score plot, (blue square among red circles, Figure 7) for a model to predict the presence of pathogenic microorganisms in the body. The patient from whom this urine sample was taken (No. 25 in Table 1) was diagnosed with lymphadenitis, and it was aseptic according to microbiological studies. However, in the results of a blood test from this patient, an elevated erythrocyte sedimentation rate was observed, which indicated the presence of inflammatory processes in the body, with possible bacterial damage to other organs. Therefore, this sample was not excluded from the sample.

The most important variables for predicting the presence of microorganisms in the source of inflammation are the signals of the sensor with the MCNT film (absolute analytical signal and the area of the “visual print”), which is sensitive to light base gases (ammonia, amines), as the most specific markers of bacterial inflammation. It has been established that there is an overestimation of the prediction results; therefore, the results are false positives for samples with the presence of mucus and fungi in the urine.

The resulting models can be used to analyze urine samples as a screening. When using a calculated indicator of infection (*InfI*), based on the results of comparing its values and clinical indicators, the sensitivity was 96%, the specificity was 50%. The relatively low specificity may be due to the lack of a complete clinical picture and patient history. So, sample No. 23 was taken from a patient with a diagnosis of closed epiphysiolysis of the right femur, and was classified as a false positive according to *InfI*. However, the patient was subsequently transferred to the purulent-septic department for surgery and isolation of *Staphylococcus haemolyticus* from the source of inflammation (sample No. 86), which confirmed the correctness of the assessment of the presence of a bacterial infection in the body according to *InfI*.

Thus, based on the results of the analysis of the EGP of urine samples with an array of sensors and a simple calculation of the infection indicator, it is possible to quickly establish the presence of bacterial diseases in children with a sufficiently high sensitivity and specificity. 

## 5. Conclusions

The work involved identifying substance markers of pathogenic processes in the EGP of urine samples by the parameters of the array of sensors. Correlations were given between the presence of substances and types of pathologies (department of the hospital). The most informative output data of an array of sensors for diagnosing bacterial pathology was established. The possibility of predicting the presence of a bacterial infection based on the results of the analysis of the equilibrium gas phase of urine samples with an array of sensors with selective film coatings was positively assessed. There are several ways to improve the metrological characteristics of PLS models using sensors arrays. One way could be to use the features of sorption kinetics of the EGP of a urine sample as numeric parameters. Another way is the consideration of the multiplication of sensor signals as variables of the model. The specificity of the way to identify bacterial diseases could be improved by a more accurate selection of parameters for calculation after analysis of a bigger sample. The use of an array of sensors in addition to traditional methods of analysis enables a quick assessment of the state of the body and the adjustment of the treatment tactics. 

## Figures and Tables

**Figure 1 sensors-22-08496-f001:**
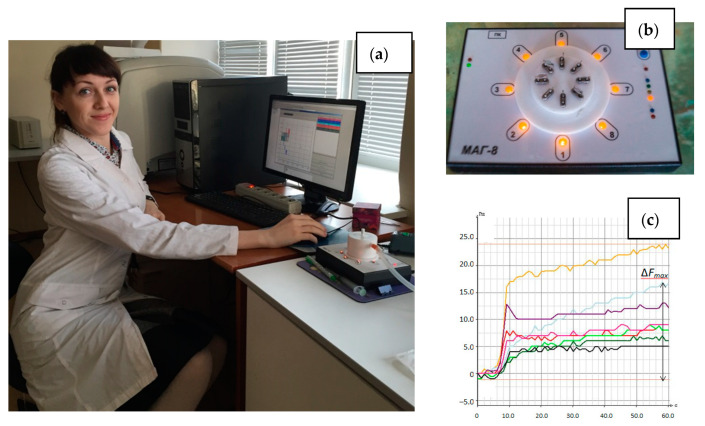
Photo of the MAG-8 device: (**a**) in the laboratory during analysis, (**b**) location of sensors in the detection cell; (**c**) chronofrequency grams of sensors during measurement.

**Figure 2 sensors-22-08496-f002:**
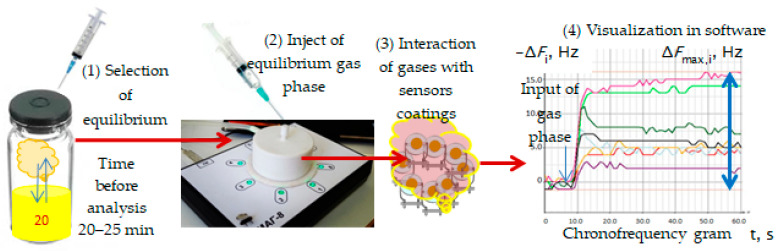
Process of measurement of gas phase of urine sample using sensor array.

**Figure 3 sensors-22-08496-f003:**
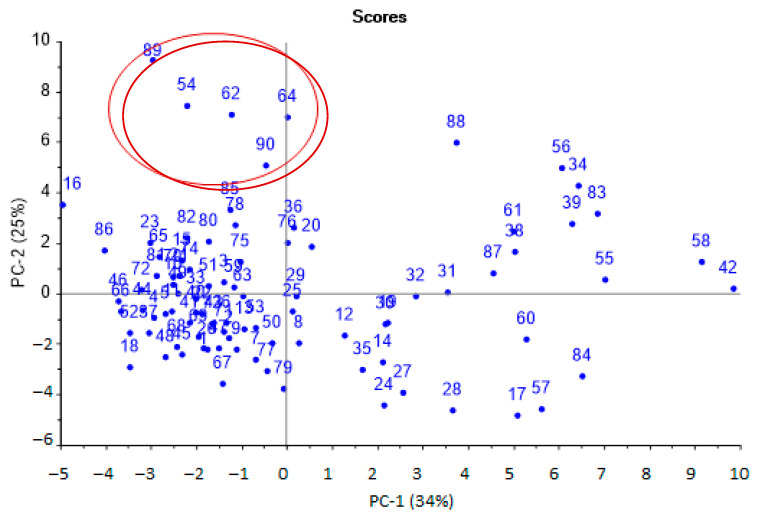
Scores plot of the PCA model for urine samples by identification parameters (sample numbers correspond to the number in Table 1,the plot with high resolution in Appendix A).

**Figure 4 sensors-22-08496-f004:**
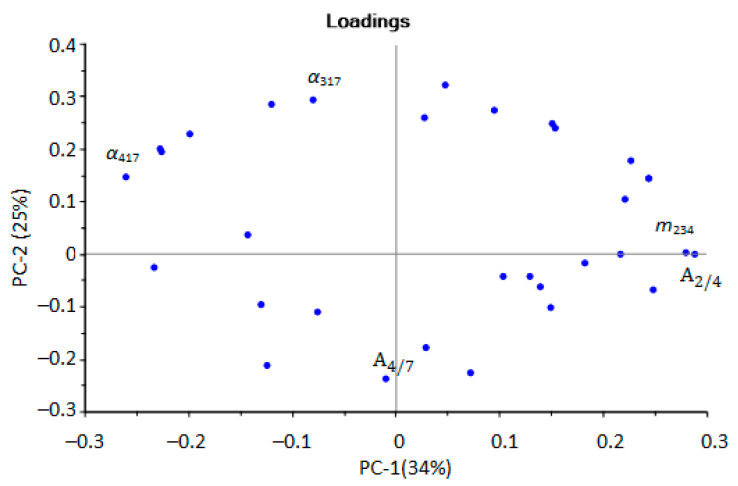
Loadings plot of the PCA model for urine samples by identification parameters (sensor numbers in the subscript correspond to the designations in the Section 2.2.1, the plot with designation of all points in Appendix A).

**Figure 5 sensors-22-08496-f005:**
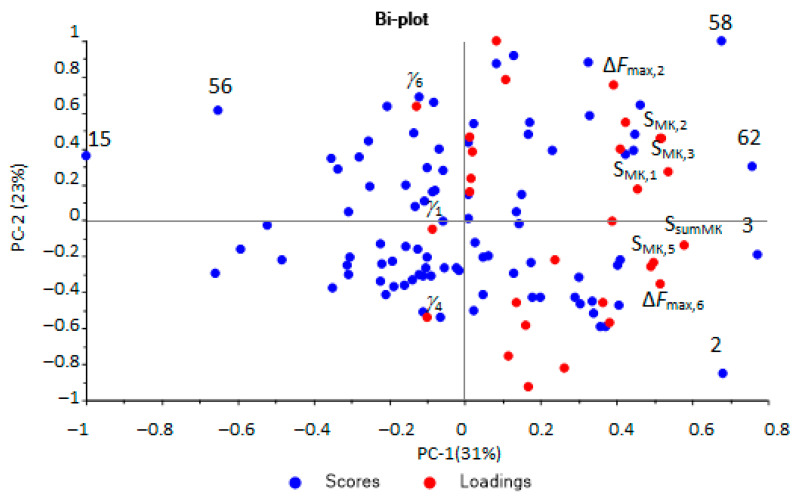
Scores and loadings plot of the PCA model for urine samples according to the optimal output data of the array of sensors (the plot with designation of all points in Appendix A).

**Figure 6 sensors-22-08496-f006:**
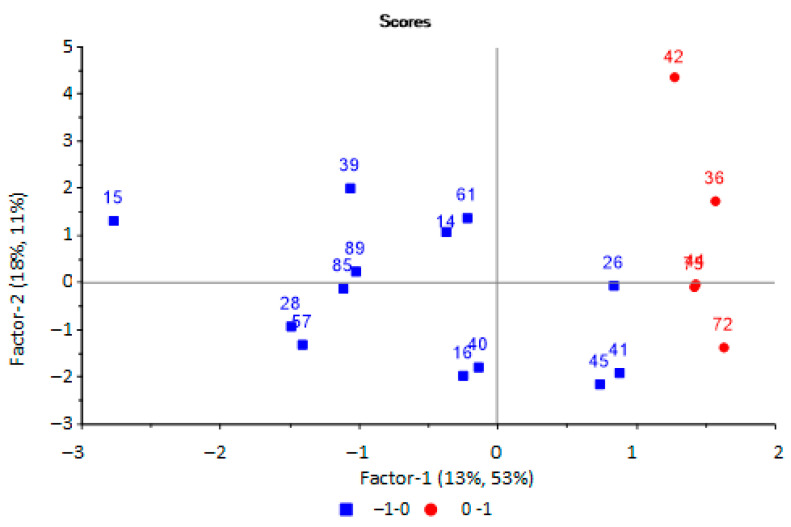
Scores plot of the PLS model for predicting the microbiological contamination of urine samples based on the results of analysis of samples by an array of sensors.

**Figure 7 sensors-22-08496-f007:**
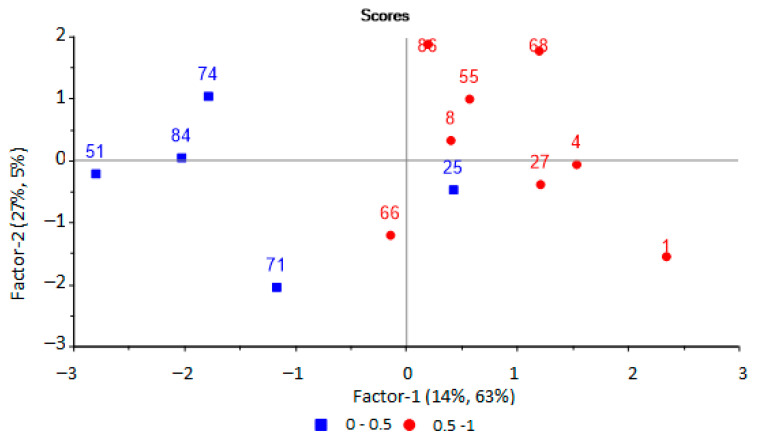
PLS-model scores plot for predicting the presence of microorganisms in biomaterial based on the results of the analysis of EGP of urine samples by an array of sensors.

**Figure 8 sensors-22-08496-f008:**
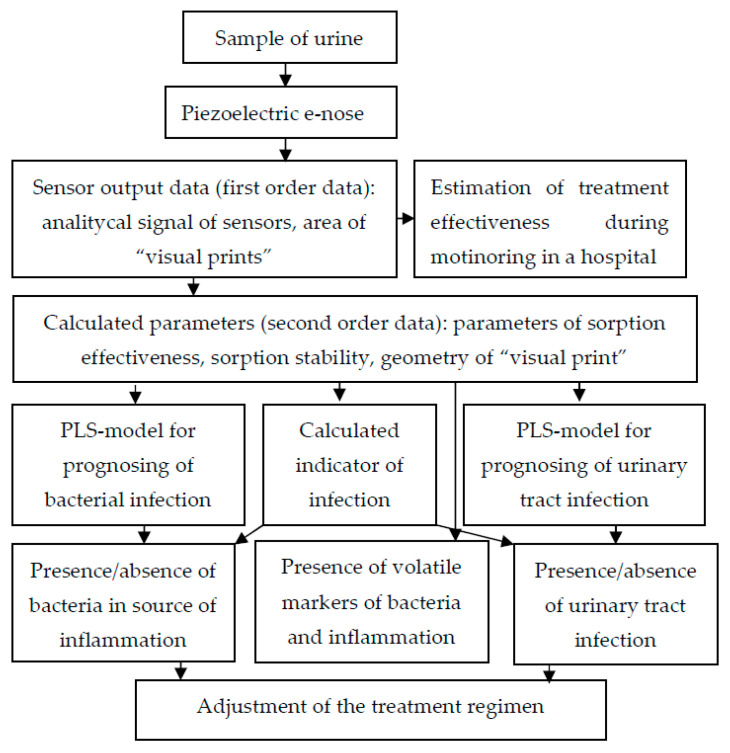
Scheme of application of piezoelectric “electronic nose” in urine analysis.

**Table 1 sensors-22-08496-t001:** Some indicators of general urine analysis (GUA) of patients and results of bacteriological investigation of biomaterial after surgery.

No.	Indicators of GUA			Isolated Microorganism from Specimen for Biopsy
	Mucus	Leukocytes	Bacteria	
1	~0	2	~0	*Staphylococcus aureus*
2	+	9	~0	–
3	~0	2	~0	– ^1^
4	~0	2	~0	*Streptococcus b-haemophylus; Candida albicans*
5	~0	2	~0	–
6	~0	2	~0	*E. coli, Streptococcus b-haemophylus*
7	~0	16	~0	*E. coli*
8	~0	3	~0	*Staphylococcus aureus*
9	~0	4	~0	–
10	~0	7	+	–
11	~0	3	~0	*Staphylococcus aureus*
12	~0	3	~0	–
13	+	2	~0	–
14	~0	4	~0	–
15	~0	16	+	–
16	~0	2	~0	–
17	~0	4	~0	–
18	~0	3	~0	–
19	~0	4	~0	–
20	~0	5	~0	–
21	++	8	+	–
22	~0	6	~0	–
23	~0	5	~0	–
24	~0	4	~0	–
25	~0	2	~0	Not determined
26	++	8	~0	–
27	~0	2	–	*E. coli*
28	~0	2	–	–
29	+	9	~0	*Staphylococcus gallinarum*
30	++	1	~0	–
31	~0	4	+	–
32	~0	6	+	–
33	+	2	+	–
34	+	2	+	–
35	+++	10	+	–
36	+++	1	+	*Staphylococcus aureus*
37	~0	1	~0	–
38	~0	1	~0	–
39	+	1	~0	–
40	~0	3	~0	–
41	+	2	~0	–
42	++	10	+	–
43	~0	1	~0	–
44	+	1	+	–
45	~0	1	~0	–
46	+++	1	+	–
47	~0	2	~0	–
48	+	3	++	–
49	+	2	+	–
50	+	2	~0	–
51	++	10	~0	Not determined
52	++	7	~0	–
53	+	3	~0	–
54	+	3	~0	*Staphylococcus aureus*
55	+	4	++	*Staphylococcus saprophysicus*
56	++	2	~0	–
57	++	1	~0	*Staphylococcus aureus*
58	++	1	~0	*Streptococcus jaccicum*
59	++	1	+	–
60	+	25	+	–
61	~0	1	~0	–
62	~0	1	~0	–
63	~0	2	~0	–
64	+	2	+	–
65	~0	1	~0	*Streptococcus viridans*
66	++	1	~0	*Streptococcus b-haemophylus*
67	++	20	+++	–
68	+	1	~0	*Staphylococcus aureus*
69	+	1	~0	–
70	+++	4	+	–
71	~0	7	~0	Not determined
72	+++	4	++	Not determined
73	~0	1	~0	–
74	++	4	+	Not determined
75	~0	1	++	Not determined
76	++	1	+	–
77	++	4	~0	–
78	+++	6	+	–
79	++	1	+	–
80	+	2	~0	–
81	+++	1	~0	*Staphylococcus epidermidis*
82	~0	3	~0	*Streptococcus b-haemophylus*
83	++	1	~0	*Klebsiella pneumoniae*
84	++	5	+	Not determined
85	~0	2	~0	–
86	+	5	~0	*Staphylococcus haemolyticus*
87	++	1	~0	*Streptococcus viridans*
88	+++	3	++	–
89	+++	2	+	–
90	+	4	~0	–

^1^ not applicable.

**Table 2 sensors-22-08496-t002:** Mass sensitivity (S_m_, Hz⋅m^3^/g) of selected sensor to VOC and reproducibility of sensor signals in their vapors.

Coating	Ammonia	Diethylamine	Butyric Acid	Ethanol	Acetone	S_r_*
PEGSb	5.00	7.82	15.0	7.72	1.41	0.04–0.13
TX-100	4.37	29.4	27.5	1.43	1.44	0.03–0.12
18C6	6.17	1.90	77.5	0.98	1.27	0.12–0.20
Tween	2.51	1.52	71.3	1.52	1.21	0.05–0.15
MR	13.4	4.00	6.72	1.39	0.65	0.02–0.17
BCB	16.4	10.2	7.12	1.16	0.83	0.05–0.20
MCNT	14.3	3.74	7.52	0.98	0.39	0.07–0.17

S_r_*—relative standard deviation of sensor signals for the studied volatile compounds.

**Table 3 sensors-22-08496-t003:** Identified substances in the EGP of urine samples from patients in departments of the children’s hospital (% of the total number of samples).

Substances	Maxillofacial Surgery	General Surgery	Traumatology	Burn	Neurosurgery	Purulent-Septic	Orthopedic
Ethanol	85	100	83	100	100	100	75
Buthanol-1	85	100	83	100	100	100	75
Acetone	32	39	22	33	83	75	50
Hydrogen sulfide	15	22	13	–	–	–	25
Phenol	76	72	56	66	83	75	50
Ethyl acetate	20	28	17	33	83	75	50
Acetic acid	76	94	83	100	100	100	75
Butyric acid	76	94	83	100	100	100	75
Dimethylacetal dimethylformamide	76	89	70	–	100	50	50
Piperidine	89	11	39	33	–	25	–
Diethylamine	70	67	39	66	50	100	50
Ammonia	– *	67	52	100	33	100	100
Amines	100	67	52	100	33	100	75
Valeric acid	50	61	52	33	100	75	50
Isovaleric acid	50	61	52	33	100	75	50

* not identified.

**Table 4 sensors-22-08496-t004:** Results of prediction of microbiological contamination of urine samples based on the results of analysis of samples with an array of sensors.

Sample No.	Predicted Value	Deviation	Reference Value
5	−0.21	0.47	−1
10	0.29	0.47	1
20	−0.29	0.57	−1
**21**	−1.38	0.64	1
34	0.15	0.68	1
**46**	−0.56	0.52	1
48	0.26	0.65	1
54	−1.55	0.78	−1
55	0.14	0.62	1
**71**	0.24	0.80	−1
79	1.17	0.66	1
80	−0.78	0.40	−1
85	−1.04	0.71	−1
90	−1.14	0.44	−1

Note: the numbers of samples with false prediction results are highlighted in bold.

**Table 5 sensors-22-08496-t005:** Results of predicting the presence of microorganisms in biomaterial based on the results of the analysis of EGP of urine samples using an array of sensors.

Sample No.	Predicted Value	Deviation	Reference Value
36	0.94	0.44	1
57	0.55	0.35	1
58	0.56	0.38	1
71	0.34	0.32	0
**75**	0.70	0.50	0
83	0.57	0.31	1

Note: the numbers of samples with false prediction results are highlighted in bold.

**Table 6 sensors-22-08496-t006:** Calculated value of infection indicator (*InfI*) for urine samples.

Sample No.	*InfI*	No.	*InfI*	No.	*InfI*	No.	*InfI*	No.	*InfI*
1	1.48	19	1.45	**37**	1.57	55	1.84	73	1.42
2	1.42	**20**	1.51	**38**	2.09	56	1.45	74	1.60
**3**	1.59	21	1.69	**39**	1.99	57	1.94	**75**	1.45
4	1.60	22	1.43	**40**	1.52	58	2.03	76	1.78
**5**	1.53	**23**	1.56	**41**	1.47	59	1.81	77	1.22
6	1.59	24	1.29	42	1.88	60	1.98	78	1.68
7	1.58	**25**	1.50	**43**	1.61	**61**	1.87	**79**	1.14
8	1.48	26	1.42	44	1.54	**62**	1.80	**80**	1.78
9	1.32	27	1.64	**45**	1.56	63	1.45	81	1.59
10	1.57	**28**	1.49	46	1.57	64	2.10	82	1.81
11	1.54	29	1.57	**47**	1.58	65	1.59	83	1.72
12	1.42	30	1.26	48	1.50	66	1.63	84	1.95
**13**	1.50	31	1.51	49	1.87	**67**	1.34	**85**	1.62
14	1.26	32	1.55	50	1.42	68	1.57	86	1.78
15	1.99	33	1.70	51	1.45	**69**	1.54	87	1.89
16	1.45	34	2.45	**52**	1.61	70	1.72	88	1.95
**17**	1.52	35	1.63	53	1.28	71	1.42	89	1.61
**18**	1.73	36	1.75	54	1.79	72	1.63	**90**	1.67

Note: Bold indicates sample numbers that are incorrectly assigned to diagnostic groups.

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
