# Peer review of "Noninvasive Detection of Bacterial Infection in Children Using Piezoelectric E-Nose"

_sensors, 2022, doi:10.3390/s22218496_

Round 1
Reviewer 1 Report
1. The description of the sensor measurement process in the manuscript should be visualized by drawing.
2. For the second part of the manuscript "Materials and Methods", the preparation of materials and sensors should be described first, followed by a description of subsequent studies.
3. Some sentences should not capitalize the first letter of a word. Page 16"So Sample № 23 was taken from a patient with a diagnosis…" The first letter of "Sample" should not be capitalized. There are many similar issues in the manuscript, please revise.
4. Please check and improve the English writing in the text.
Author Response
Response to Reviewer 1 Comments
The authors are grateful to the Reviewer for a detailed study of the manuscript and valuable comments. The followîng improvements are made:
Point 1: The description of the sensor measurement process in the manuscript should be visualized by drawing.
Response 1: A drawing of the measurement process using an array of sensors has been added to the Materials and Methods section (Figure 2).
Point 2. For the second part of the manuscript "Materials and Methods", the preparation of materials and sensors should be described first, followed by a description of subsequent studies.
Response 2: The explanations about the preparation of materials and sensors during the formation of coatings includes in the Materials and Methods section.
Point 3: Some sentences should not capitalize the first letter of a word. Page 16 "So Sample No. 23 was taken from a patient with a diagnosis…" The first letter of "Sample" should not be capitalized. There are many similar issues in the manuscript, please revise.
Response 3: The text of the manuscript was checked and the misprints were corrected.
Point 4: Please check and improve the English writing in the text.
Response 4: The text of the manuscript in English has been checked and corrected.
Reviewer 2 Report
The paper shows using an electronic nose to detection of bacterial infection in children and discuss its errors in detail. However, I cannot recommend to accept this article in current version unless the authors address the following concerns:
1. The methods and materials do not provide the analytical characteristics of the selected sensors, such as sensitivity to compound vapors, the limit of detection of volatile compounds in the gas phase over urine samples.
2. In the text of the article, there is little discussion of the set of variables from the array of sensors, which is used to construct PLS models after optimisation. This process should be described in more detail. The original or optimized sensor dataset used to build the models provided in the supplement materials would also be helpful.
3. In the results, the characteristics of the obtained PLS models are not fully shown, for example, R2, slope, explained variance. Perhaps, on the basis of this, in conclusion, it would be possible to suggest ways to increase the sensitivity and specificity of detecting a bacterial infection for the supposed approach?
Author Response
Response to Reviewer 2 Comments
The authors are grateful to the reviewer for a detailed study of the manuscript and valuable comments. The following improvements are made:
Point 1: The methods and materials do not provide the analytical characteristics of the selected sensors, such as sensitivity to compound vapors, the limit of detection of volatile compounds in the gas phase over urine samples.
Response 1: The analytical characteristics of the sensors have been added to the Methods and Materials section in the form of Table No. 2 and in the text of subsection 2.2.1
Point 2: In the text of the article, there is little discussion of the set of variables from the array of sensors, which is used to construct PLS models after optimisation. This process should be described in more detail. The original or optimized sensor dataset used to build the models provided in the supplement materials would also be helpful.
Response 2: The resulting optimized sets of sensor parameters for building PLC models are described in the text of the article in the Results section. The optimized sensor datasets used to build the models are provided in attachment.
Point 3: In the results, the characteristics of the obtained PLS models are not fully shown, for example, R2, slope, explained variance. Perhaps, on the basis of this, in conclusion, it would be possible to suggest ways to increase the sensitivity and specificity of detecting a bacterial infection for the supposed approach?
Response 3: Characteristics of the obtained PLS models were added to the text of the manuscript in the results section. In conclusion, ways are proposed to increase the sensitivity and specificity of detecting a bacterial infection.

Reviewer 3 Report
Kuchmenko et al work on Noninvasive detection of bacterial infection in children using piezoelectric e-nose is interesting and useful for the scientific community. However, before being accepted, the following comments should be addressed.
Comments
1. Abstract is too lengthy, reduce it.
2. English language should be polished throughout the manuscript.
3. What is the detection limit of the sensor?
4. What is the reproducibility of sensor
Author Response
Response to Reviewer 3 Comments
The authors are grateful to the reviewer for a detailed study of the manuscript and valuable comments. The comments about suggestions:
Point 1: Abstract is too long, reduce it.
Response 1: The abstract of the article has been reduced.
Point 2: English language should be polished throughout the manuscript.
Response 2: The text of the manuscript in English has been checked and corrected.
Point 3: What is the detection limit of the sensor?
Response 3: The information about limits of detection for VOCs are added in the Materials and Methods section with relative references.
Point 4: What is the reproducibility of sensor
Response 4: The relative standard deviations of sensor signals are added in the Materials and Methods section in Table 2 as reproducibility charcteristic of sensor.